# An enhanced intracellular delivery platform based on a distant diphtheria toxin homolog that evades pre-existing antitoxin antibodies

Shivneet K Gill [1,2], Seiji N Sugiman-Marangos[2], Greg L Beilhartz[2], Elizabeth Mei [3,4], Mikko Taipale [3,4] & Roman A Melnyk [1,2]✉

## Abstract

Targeted intracellular delivery of therapeutic proteins remains a significant unmet challenge in biotechnology. A promising approach is to leverage the intrinsic capabilities of bacterial toxins like diphtheria toxin (DT) to deliver a potent cytotoxic enzyme into cells with an associated membrane translocation moiety. Despite showing promising clinical efficacy, widespread deployment of DT-based therapeutics is complicated by the prevalence of pre-existing antibodies in the general population arising from childhood DT toxoid vaccinations, which impact the exposure, efficacy, and safety of these potent molecules. Here, we describe the discovery and characterization of a distant DT homolog from the ancient reptile pathogen *Austwickia chelonae* that we have dubbed chelona toxin (ACT). We show that ACT is comparable to DT structure and function in all respects except that it is not recognized by pre-existing anti-DT antibodies circulating in human sera. Furthermore, we demonstrate that ACT delivers heterologous therapeutic cargos into target cells more efficiently than DT. Our findings highlight ACT as a promising new chassis for building next-generation immunotoxins and targeted delivery platforms with improved pharmacokinetic and pharmacodynamic properties.

**Keywords** Immunotoxin; Intracellular Delivery; Diphtheria Toxin; Chelona Toxin; Antidrug Antibodies
**Subject Categories** Biotechnology & Synthetic Biology; Methods & Resources; Microbiology, Virology & Host Pathogen Interaction

## Introduction

Antibody drug conjugates (ADCs) have emerged as one of the fastest growing classes of targeted cancer therapies, with over a dozen FDA approvals in the past decade (Dumontet et al, 2023). Immunotoxins are a subclass of ADCs with a storied history that date back to Paul Ehrlich's original "magic bullet" hypothesis (Chaudhary et al, 1987; Moolten & Cooperband, 1970; ROSS et al, 1980; Vollmar et al, 1987). Immunotoxins act by targeting cancer receptors via a receptor binding domain but, in contrast to more traditional ADCs which release a toxic small molecule into cells, immunotoxins deliver a cytotoxic protein enzyme into cells. Unlike the toxic small molecule payloads of ADCs that after being released within the cancer cell can kill neighboring "healthy" cells through a process known as the bystander effect, the delivered cytotoxic enzyme payloads of immunotoxins are unable to freely diffuse into neighboring receptor-negative cells (Hassan et al, 2016). Immuno-toxins leverage the architecture of bacterial exotoxins which contain a receptor binding domain (R) capable of binding host cells (Mitamura et al, 1995), a translocation domain (T) that forms a pore in endosomal membranes upon receptor binding and internalization of the toxin (Oh et al, 1999), and a cytotoxic domain (C) encoding a highly processive enzyme that is delivered to the cytosol to target and inactivate an important intracellular protein to cause cell death (Strauss & Hendee 1959) (Fig. 1A).

To date there have been three FDA approved immunotoxins: denileukin diftitox, (Ontak™) (U.S. Food and Drug Administration, 1999); tagraxofusp, (Elzonris™) (U.S. Food and Drug Administration, 2018a); and moxetumomab pasudotox, (Lumoxiti™) (U.S. Food and Drug Administration, 2018b) of which the first two are derivatives of diphtheria toxin (DT). DT is a highly toxic protein secreted by *Corynebacterium diphtheriae*, and the etiologic agent of the disease Diphtheria (Roux and Yersin, 1888). Denileukin diftitox and tagraxofusp contain the catalytic and translocation domains of DT, but the receptor binding domain has been replaced with either interleukin-2 (IL-2) or interleukin-3 (IL-3), respectively, which successfully re-targets DT to kill cells expressing their receptors CD25 (IL2RA) or CD123 (IL3RA). As such, Ontak™ was approved for cutaneous T cell lymphoma (CTCL), and Elzonris™ was approved for blastic plasmacytoid dendritic cell neoplasm (BPDCN), increasing survival rates by over 50% for BPDCN patients. Elzonris™ is currently being explored for patients with acute myeloid leukemia (AML) (2024), as AML cells also express high levels of CD123 (Testa et al, 2002).

To stop the devastating spread of diphtheria in the 20th century, a DT toxoid vaccine was developed in the early 1920s that was rapidly

[1]Department of Biochemistry, University of Toronto, Toronto, ON M5S1A8, Canada. [2]Molecular Medicine Program, The Hospital for Sick Children Research Institute, 686 Bay Street, Toronto, ON M5G 0A4, Canada. [3]Department of Molecular Genetics, University of Toronto, Toronto, ON M5S1A8, Canada. [4]Donnelly Centre for Cellular and Biomolecular Research, University of Toronto, Toronto, ON M5S 3E1, Canada. ✉E-mail: roman.melnyk@sickkids.ca

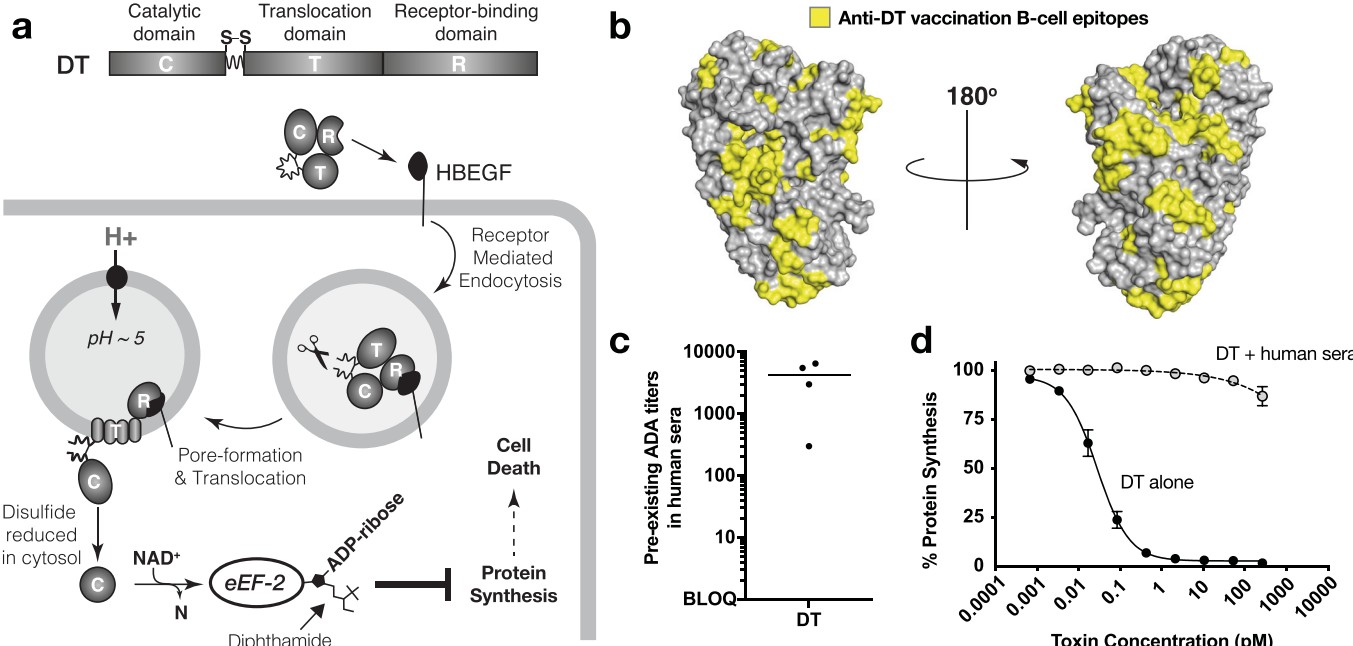

**Figure 1. Diphtheria toxin (DT) mechanism-of-action and impact of pre-existing anti-DT antibodies.**

(A) Molecular mechanism of DT intoxication pathway. DTs R-domain (DT$_R$) binds HBEGF and undergoes HBEGF-mediated internalization. Furin protease cleaves at a site between the C- and T-domains, and as the endosome acidifies, the DT translocase (DT$_T$) unfolds and inserts into the endosomal membrane to form a pore, through which the catalytic domain (DT$_C$) translocates through and enters the cytosol. Disulfide reduction releases DT$_C$ into the cytosol, where it ADP-ribosylates elongation factor 2 (EF-2), inhibiting its function in mediating ribosomal protein translation, thus shutting down protein synthesis in the cell to cause cell death. (B) B-cell epitopes on DT (yellow) because of vaccination, as characterized by De-Simone et al. (C) Assessment of IgG binding to DT from pooled human sera, by ELISA. $n = 4$ (technical replicates), SEM. (D) Human sera neutralization of DT, on cell-based assay. Sera is neutralizing beyond 100 pM. $n = 4$ (biological replicates), SEM. Source data are available online for this figure.

adopted and has become a mainstay in childhood vaccination programs; currently ~85% of the global population is vaccinated against DT (Kaur et al, 2023). Vaccine-induced antibodies recognize and bind epitopes on DT and prevent host cell intoxication, thus providing protection against the devastating symptoms of diphtherial disease. Anti-DT antibodies found widely throughout the human population, however, also pose a barrier to the broad application of DT-based treatments. Indeed, clinical trial data for both Ontak™ and Elzonris™ highlight the problematic negative impact of vaccine-induced pre-existing anti-drug antibodies (ADAs) on an otherwise highly effective targeted therapeutic (U.S. Food and Drug Administration, 1999, 2018a). In total, 66% of CTCL patients had baseline ADAs to Ontak™ of which 45% were neutralizing (U.S. Food and Drug Administration, 1999), and up to 95% of BPDCN patients had baseline ADAs to Elzonris™ prior to treatment (Jen et al, 2020). Strikingly, there was a clear inverse relationship between pre-existing ADA titers and drug exposure; in patients with the highest pre-existing anti-DT titers, drug exposure was up to 100-fold lower than in patients with low titers (PRZEPIORKA et al, 2018). Moreover, patients with pre-existing ADAs had lower response rates. Finally, patients with pre-existing antitoxin antibodies can develop rapid anamnestic high-titer antibody responses after treatment with the associated immunotoxin (Wayne et al, 2014).

Devising strategies to evade pre-existing ADAs would improve all aspects of DT-based therapeutics from pharmacokinetics to pharmacodynamics, ultimately resulting in a better, more durable outcome for cancer patients (Kreitman, 2012). As part of our on-going interest in understanding the evolutionary origins and molecular ancestry of bacterial toxins implicated in human diseases (Mansfield et al, 2018; Orrell et al, 2020; Sugiman-Marangos et al, 2022a), in this study we characterized distant homologs of DT; however, in this case with the specific goal of identifying a toxin scaffold or scaffolds that retained DT functionality but are not recognized by pre-existing DT ADAs. Our identification of two related toxins from the reptile-associated pathogen *Austwickia chelona* that satisfied these criteria led to the discovery of a novel chassis for targeted cancer therapeutics and protein delivery.

# Results

## Functional screening and characterization of diphtheria toxin homologs

Vaccine-related antibody epitopes are distributed across all three domains and cover much of the surface of DT (De-Simone et al, 2021) as illustrated in Fig. 1B. To demonstrate the impact that circulating anti-DT antibodies have on DT functional intoxication, we first confirmed the presence of anti-DT antibodies in pooled human sera using a DT-based ELISA (Fig. 1C). Mixing human sera with purified DT resulted in a dramatic reduction in the cytotoxicity of DT by more than six orders of magnitude (Fig. 1D), illustrating at a functional level how prior vaccination provides neutralizing protection from DT intoxication. At the same time these data highlight the scale of the barriers facing

would-be DT therapeutics and the rationale for searching for strategies to circumvent pre-existing ADA.

In an attempt to identify a candidate toxin scaffold that evaded pre-existing antitoxin ADA, we characterized eight representative members of the known extended DT toxin family we identified previously through bioinformatics (Mansfield et al, 2018; Sugiman-Marangos et al, 2022a). The highly related DT variants from *C. ulcerans* and *C. pseudotuberculosis* were excluded from analysis owing to their extremely high sequence identity (i.e., 95–99%). The remaining DT-like putative gene sequences all shared between 20 and 40% sequence identity to DT (Appendix Fig. S1; Fig. 2A). The functionality of the individual domains from the DT homologs was evaluated using a modular "host-guest" strategy, in which the translocase or catalytic domains from DT (i.e., the host) were replaced with the corresponding domains from a given homolog (i.e., the guest) and tested for functional intoxication (Fig. 2B,C). The native DT receptor binding moiety (i.e., $DT_R$) was used in all constructs to standardize the receptor-binding step in the intoxication pathway (Fig. 1A). As a universal readout of function, we quantified inhibition of protein synthesis by the delivered cytotoxic ADP-RT in Vero cells engineered to constitutively express a destabilized luciferase (Vero-NLucP).

Only three of the translocases from distant DT homologs displayed the capacity to deliver $DT_C$ into cells (Fig. 2B). The two most active translocases were both derived from toxins produced by different species of *Austwickia chelonae*. Notably, these strains have been reported to cause skin lesions in various reptiles (bearded dragon (Tamukai et al, 2016), king cobra (Wellehan et al, 2004), alligators, turtles (MASTERS et al, 1995), and tortoises), and specifically the strain LK16-18 has been reported to cause cutaneous granuloma in crocodile lizards (Jiang et al, 2019). A similar host-guest strategy undertaken with the catalytic domains revealed that four of the seven homologs, including those from *Austwickia* were functional to varying degrees (Fig. 2C). From these analyses, the two homologs from *Austwickia chelonae* emerged as promising candidates to pursue given that they were the only to show both significant translocase and catalytic activity. Hereafter, we refer to the protein from *A. chelonae* as Chelona Toxin 1 (ACT1) and *A. chelonae LK16-18* as Chelona Toxin 2 (ACT2).

## Structural and functional characterization of chelona toxins

To better characterize the structure and function of ACT1 and ACT2, we next sought to elucidate their three-dimensional structures. ACT1 was crystallized using hanging drop vapor diffusion and the X-ray crystal structure was determined at 2.50 Å resolution (Fig. 2D). For ACT2, we used AlphaFold 2.0 (Jumper et al, 2021; Mirdita et al, 2022) to model its three-dimensional structure (Fig. 2D; Appendix Fig. S2). The overall structures, the relative orientations of their individual domains and the secondary structural motifs in ACT1 and ACT2 were highly similar to one another and to DT with a root mean square deviation (RMSD) of 2.3 Å to DT for ACT1 (401/535 residues aligned) and 1.3 Å to DT for ACT2 (383/535 residues aligned).

Despite having functional catalytic and translocation domains and highly similar structures (Fig. 2B,C), ACT1 and ACT2 both displayed minimal toxicity toward mammalian cells (Appendix Fig. S3). We posited that the lower-than-expected potency for

ACT1 and ACT2 on Vero cells, which express high levels of the DT receptor HBEGF, was due to ACT1 and ACT2 not using HBEGF as a receptor to enter cells. Indeed, the residues implicated in binding to HBEGF in $DT_R$ are poorly conserved in the corresponding sites in ACT1 and ACT2 (Appendix Fig. S4). To experimentally determine whether $ACT1_R$ and $ACT2_R$ bind a cell-surface receptor on human cells, we conducted a genome-wide CRISPR/Cas9 screen on Hap1 cells with the TKOv3 library. Chimeras were made in which the R-domains of ACT1 and ACT2 were recombinantly linked to the catalytic and translocation domains of DT and tested on Hap1 cells (Fig. 2E,F). The top hits in the ACT1 CRISPR/Cas9 screen were genes implicated in diphthamide biosynthesis—the molecular target for ADP-ribosylation by $DT_C$ (Fig. 2G). Notably absent in this screen was HBEGF—the top hit in a similar screen using wildtype DT (Appendix Fig. S5). Unique among the top hits in this screen was sortilin (SORT1)—a type I membrane glycoprotein trafficking receptor (Mitok et al, 2022). SORT1 knockout cells were generated and found to be protected from DT($ACT1_R$) while remaining susceptible to DT (Fig. 2H; Appendix Fig. S6). Similarly, SORT1 overexpression through lentiviral transduction re-sensitized these cells to DT($ACT1_R$) (Fig. 2H). These data suggest that SORT1 is essential for ACT1 toxicity, likely serving as a cell-surface receptor to mediate ACT1 entry into cells.

## Chelona toxin is not recognized by pre-existing anti-DT antibodies in human sera

Next, we evaluated whether ACT was recognized by anti-DT antibodies present in human donor sera. Since ~50% of the residues within the specific B-cell epitopes on DT differ in equivalent sites in ACT1 and ACT2 (Fig. 3A; Appendix Fig. S7), we anticipated low cross-reactivity to ACT. To experimentally determine whether pre-existing anti-DT antibodies bind ACT we produced constructs of DT and ACT in which the native receptor-binding domains were removed and replaced with an affibody that is known to be non-immunogenic. Removal of the native receptor-binding domains, which would not be present in any therapeutic toxin design, helps guard against an overestimation of the extent of binding of pre-existing anti-DT antibodies. Nevertheless, we observed a dramatic difference in antibody titers in human sera between $DT_{1-389}X$ and $ACT_{1-389}X$; whereas the DT-based scaffold showed titers of ~$10^6$, the titers against the equivalent ACT-based scaffold were below the limit of quantification. In addition to confirming that ACT is not recognized by pre-existing anti-DT antibodies, the complete lack of binding of antibodies to ACT in human sera observed suggests that humans are likely not frequently exposed to *Austwickia chelonae* or its toxins (Fig. 3B).

## Chelona toxin-based immunotoxins are functional and evade neutralization by pre-existing ADAs

An important feature of DT that makes it a powerful immunotoxin platform is its ability to be retargeted to different receptors using a variety of binding moieties. To evaluate the potential of ACT as a replacement chassis for DT in immunotoxin designs, we constructed two different retargeted DT- and ACT-based immunotoxins. The first was to fuse an scFv targeting CD123 (Kovtun et al, 2018) to DT and ACT (Fig. 3C). DT-CD123 and ACT-CD123 were tested on MV-4-11 AML cells in the absence and presence of human sera, and cell viability was measured. While both

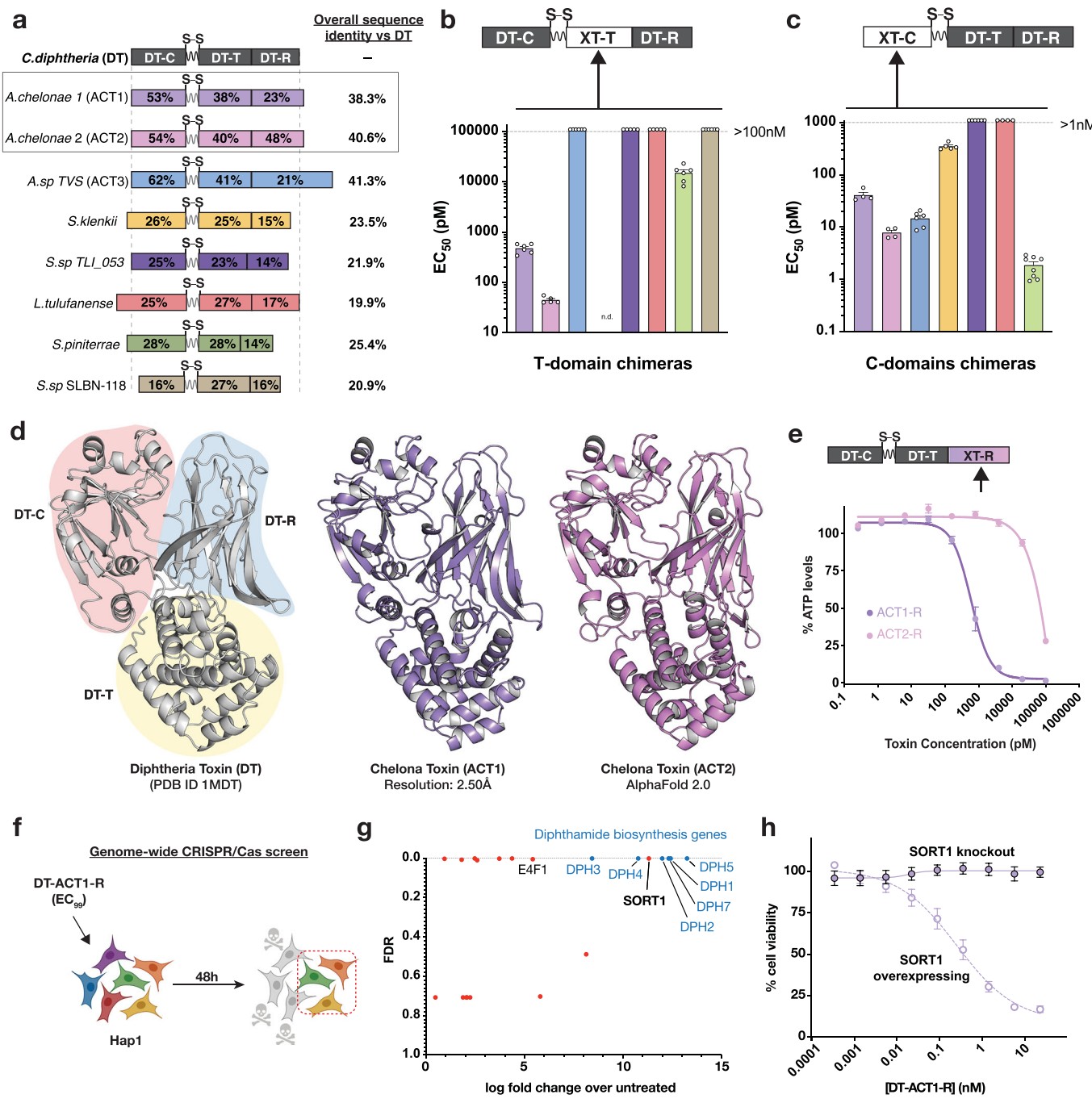

**Figure 2. Functional and structural characterization of DT homologs.**

(A) Sequence identity of putative DT-homologs to DT. Species from which DT-homologs were identified are indicated on the left. Approximate length of each sequence is depicted by the length of the bars, with DT representing 535 residues. (B, C) Screen for functional translocation domains and catalytic domains, respectively. Each chimeric protein was tested on vero-nLucP cells, and $EC_{50}$ values were plotted on the y-axis. Colors correspond to the respective DT-homolog in (A). $n = 3$ (2 technical × 3 biological replicates), SEM. The reference EC50 values for wildtype DT in (B) and (C) are 0.6 ± 0.2 pM and 0.7 ± 0.2 pM, respectively. (D) Crystal structure of ACT1 (PDB 9BIW) and AlphaFold 2.0 structure of ACT2. (E) Testing $ACT1_R$ and $ACT2_R$ functionality on human cells. Recombinant proteins were tested on Hap1 cells. $ACT1_R$ had an $EC_{99}$ of ~75 nM. (F) Schematic of the genome-wide CRISPR/Cas9 screen. (G) Results of CRISPR/Cas9 screen. (H) Validation of SORT1 as a receptor for $ACT1_R$, using Hap1-SORT1 knockout cells and SORT1 overexpressing cells. $n = 3$ (2 technical × 3 biological replicates), SEM. Source data are available online for this figure.

immunotoxins were toxic to cells, DT-CD123 was potently neutralized by sera by four orders-of-magnitude while the potency of ACT-CD123 was unaffected by sera (Fig. 3D). Similarly, we constructed dual-targeted DT- and ACT-based chimeric toxins

using a HER3 (Nazari et al, 2019) targeting affibody and a peptide targeting integrin αvβ6 (DiCara et al, 2007) (Fig. 3E). Similar to above, in the presence of human sera, the DT-based immunotoxin was significantly neutralized whereas the human sera had no effect

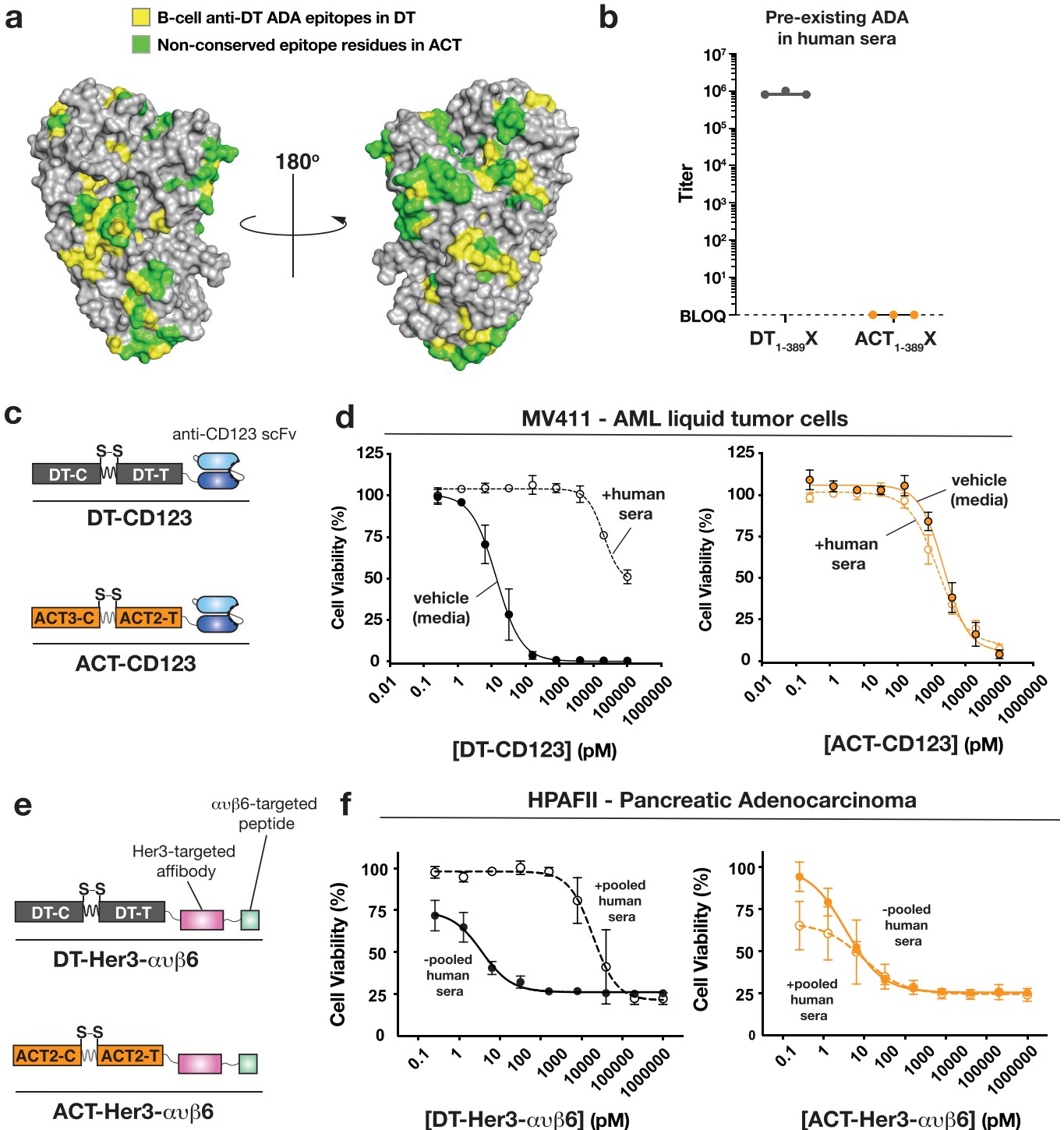

**Figure 3. Chelona toxins evade pre-existing ADA and can function as immunotoxins.**

(A) Conservation of epitopes between DT and ACT1. DT (pdb 1mdt) is used as the backbone. In yellow are epitopes conserved in ACT1, and in green are epitopes that are not conserved. (B) ELISA showing binding of human sera to either the catalytic domain and translocation domain of DT or ACT. $DT_{1-389}$ is the first 389 residues of DT which contain the C- and T-domains of DT. $ACT_{1-389}$ is the catalytic domain of *A.sp TVS* and translocation domain of ACT2. X in both cases is the scFv binding CD123. $n = 3$ (biological replicates), SEM. (C) Diagram of the DT- and ACT-based immunotoxins binding CD123. (D) CD123 binding immunotoxins tested on AML cells, in the presence and absence of human sera. DT is significantly neutralized in the presence of human sera, whereas ACT is not. $n = 3$ (biological replicates), SEM. (E) Diagram of DT- and ACT-based immunotoxins binding HER3 and integrin αvβ6. (F) Immunotoxins tested on pancreatic cancer cells, in the presence and absence of human sera. DT is significantly neutralized in the presence of human sera, whereas ACT is not. $n = 3$ (3 biological replicates), SEM. Source data are available online for this figure.

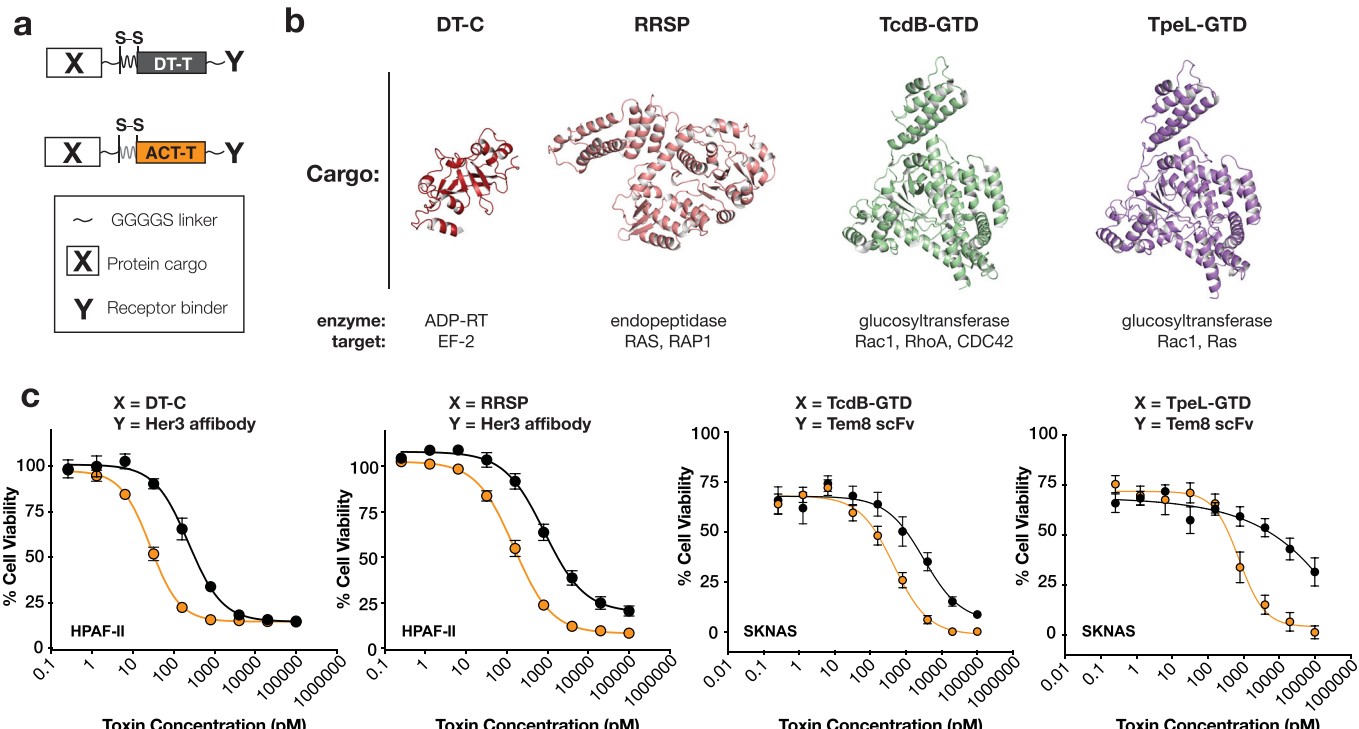

**Figure 4. Delivery of diverse heterologous cargos by ACT and DT.**

(A) Schematic of designed immunotoxins delivering diverse cargos. (B) Structures and functions of catalytic domains (cargos) delivered by either the DT or ACT translocases. PDB IDs for the structures are as follows: $DT_C$: 1mdt, RRSP: 5w6l, TcdB-GTD: 7s0z, TpeL-GTD: 9BON. (C) Cellular delivery of each cargo by the translocation domain of DT or ACT, using various receptor binding domains. $n = 3$ (2 technical × 3 biological replicates), SEM. Source data are available online for this figure.

on the potency of the ACT-based immunotoxin (Fig. 3F). These results demonstrate the profound impact that pre-existing anti-DT ADA have on the efficacy of DT-based immunotoxins and provide further support to using ACT as a viable replacement for DT in future immunotoxin designs.

## The ACT translocase is superior to DT in delivering heterologous cargos into cells

Recently, we and others, have demonstrated the potential of $DT_T$ to deliver other protein cargos of interest into cells in vitro and in vivo, thereby expanding the therapeutic utility of DT as an intracellular delivery platform (Arnold et al, 2020; Auger et al, 2015; Tian et al, 2022; Vidimar et al, 2020). To compare the ability of $ACT_T$ to serve as a general translocase of diverse protein payloads of therapeutic interest, we designed DT and ACT-based translocase constructs flanked by flexible linkers that were retargeted with either a HER3-targeting affibody (Nazari et al, 2019) or a TEM8-targeting scFv (Szot et al, 2018) (Fig. 4A,B). Four protein cargos ranging in size from 23 to 63-kDa were selected, including $DT_C$ itself; RRSP, an endopeptidase that cleaves all Ras isoforms and Rap1; and the glucosyltransferase domains (GTDs) from *C. difficile* Toxin B (TcdB-GTD), and *C. perfringens* large toxin (TpeL-GTD) that target different intracellular small GTPases. Each cargo was placed at the amino terminus of a cassette consisting of a flexible linker and the furin site flanked by cysteines that form a disulfide bond from DT (Fig. 4A). The four HER3

targeted chimeras were tested on human pancreatic HPAF-II cells (high HER3 expression), and the four TEM8-targeted chimeras were tested on human neuroblastoma SKNAS cells (high TEM8 expression) (Fig. 4C). Remarkably, in each case, $ACT_T$ was superior to $DT_T$ in delivering heterologous cargos into cells. The $ACT_T$-containing constructs were ~10–50-fold more potent than the corresponding $DT_T$-containing chimera suggesting that $ACT_T$ is a more efficient protein translocase than $DT_T$.

## The CT translocase lacks the safety latch that enables "earlier" endosomal escape of its cargo

To elucidate the basis for the enhanced delivery by $ACT_T$, we further evaluated the structure and function of the two ACT translocases relative to the well-studied DT translocase (Fig. 5A). Models for the pH-dependent unfolding and insertion of the translocation domain have been proposed for DT. In particular, glutamic acid 349, aspartic acid 352, and six histidine residues (i.e., 223, 251, 257, 322, 323, 372) have all been implicated in the various processes related to translocation (Perier et al, 2007; Rodnin et al, 2016; Rodnin et al, 2023). Glu-349 (O'Keefe et al, 1992) and Asp-352 (Ghatak et al, 2015) are thought to be important in driving and maintaining the insertion of the translocase in the endosomal membrane, and both are conserved in ACT1 and ACT2 suggesting these are not responsible for their improved translocase function (Fig. 5B). A striking difference in ACT translocases relates to the number and location of histidines relative to DT (Appendix Fig. S8). It was previously reported that the

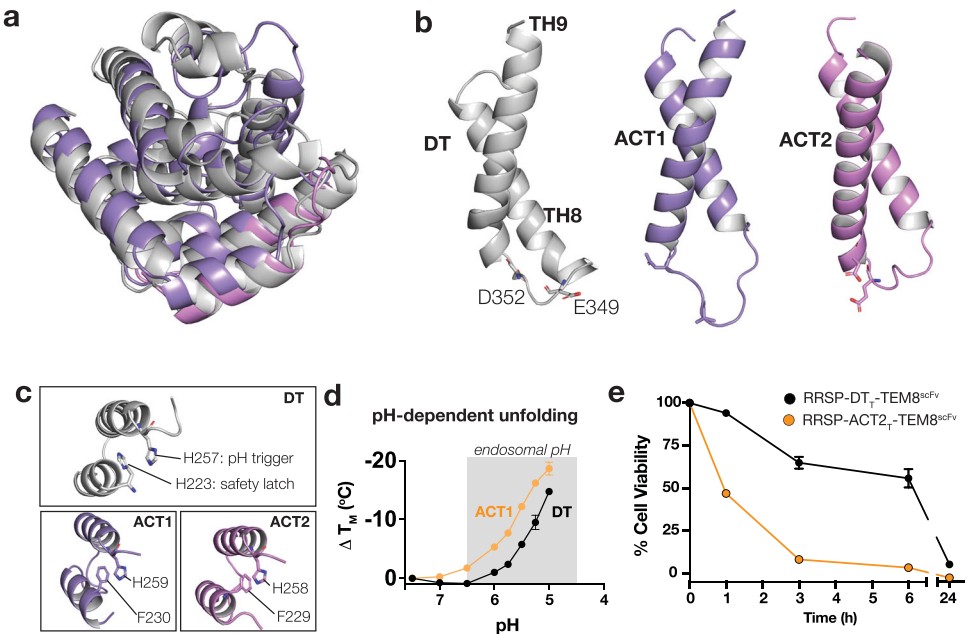

**Figure 5. Molecular insights into the chelona toxin translocase.**

(A) Structure overlay of translocases from DT (gray), ACT1 (purple), and ACT2 (pink). (B, C) Key functional residues are depicted in stick format. Colours correspond to (A). (D) Melting temperature of DT and ACT1, by differential scanning fluorimetry. DT: $n = 4$, ACT1: $n = 7$ (technical replicates), SEM. (E) Kinetics experiment on HCT116 cells. Time points indicate how long toxins were incubated with cells before replacing cells with toxin-free media. ATP levels were measured at 72 h and plotted as cell viability on the y-axis. $n = 2$ (technical replicates), SEM. Source data are available online for this figure.

unfolding of $DT_T$ at a pH <6.0 is mediated by the local effects of His-223 (i.e., the "safety latch") that lowers the pKa of His-257 (Rodnin et al, 2016). These residues are thought to control the pH-dependent unfolding of $DT_T$, which initiates the translocation process. In both ACT1 and ACT2, however, the safety latch is absent. In the position occupied by a histidine in DT is a phenylalanine for both ACT variants (Fig. 5C). Notably, it was shown previously that disrupting the safety latch in DT results in an increase in the pH at which $DT_T$ unfolds (Rodnin et al, 2016). To assess whether ACT unfolds at a higher pH than DT, we used differential scanning fluorimetry (DSF) (Sugiman-Marangos et al, 2022b) and assessed the melting temperature of DT or ACT at pH 5.0–7.5. Indeed, we observed a shift in the pH-dependent unfolding profile of ACT relative to DT, suggesting that ACT initiates the unfolding process at a higher pH than DT (Fig. 5D).

We hypothesized that the higher pH threshold for unfolding for ACT-based translocases may enable earlier, or more rapid escape from endosomes that become increasingly acidic as they transit from the cell surface toward late endosomes and lysosomes. To assess if the kinetics of translocase delivery differed, we compared the kinetics of delivery of RRSP by retargeted $DT_T$ and $ACT_T$ constructs. Each chimera was incubated with cells for 1 h, 3 h, 6 h, and 24 h, after which cells were washed and allowed to grow for 72 h before reading for viability. As shown in Fig. 5E, $ACT_T$ delivered RRSP much more rapidly into cells than $DT_T$ across all time points tested. For instance, whereas RRSP delivered by $ACT_T$ reduced the viability of cells below 50% by 1 h, cells treated with the same dose of a similar $DT_T$ construct showed greater than 60% viability even after 6 h of treatment. These data show that the ACT translocase more rapidly delivers its cargo into cells, potentially as a result of it forming pores earlier (i.e., at higher pH's) within

endosomes (Fig. 5E). We cannot exclude the possibility that other factors, including the efficiency of insertion of the $ACT_T$ pores relative to $DT_T$ pores, or the rate of translocation through the two pores may contribute to $ACT_T$ being a superior delivery apparatus to $DT_T$.

## Discussion

The work presented herein was undertaken to address the numerous liabilities associated with pre-existing ADAs to immunotoxin therapeutic platforms. The scale of the problem of pre-existing antibodies on immunotoxin therapy is emphasized by the fact that patients with pre-existing antibodies to Pseudomonas exotoxin A (PE)-based immunotoxins have been excluded from participation in PE-based immunotoxin trials (Kreitman et al, 2012). For DT-based immunotoxins, patient exclusion is problematic as most of the population has been previously vaccinated against toxoid DT during childhood. Our approach to do a functional screen of DT homologs uncovered toxins from the *Austwickia* genus as promising candidates to replace DT as a potential alternative immunotoxin platform. These novel immunotoxins are not neutralized or recognized by ADAs in human sera and retain equivalent or superior functionality to DT in their capacity as immunotoxins and protein delivery vectors. While *Austwickia chelonae* is a reptile pathogen, it has never been reported as a human pathogen and our data suggest that humans likely have not been exposed to these toxins. This is unlike PE, the other major toxin used in immunotoxins. Despite not being formally immunized against PE, many humans harbour pre-

existing ADAs arising from periodic or constant prior exposure to toxigenic *Pseudomonas aeruginosa*, which is a commensal human pathogen (U.S. Food and Drug Administration, 2018b).

Solid tumors remain the most difficult cancers to treat using immunotoxins as these patients typically have intact immune systems and thus are capable of mounting more substantial immune responses than patients with hematologic malignanices (Mazor et al, 2016). In this study we showed that ACT can be retargeted to HER3, integrin αvβ6, and TEM8, which are highly expressed in a number of solid tumors. An intriguing finding of this study was that ACT uses SORT1 as a cell-entry receptor for intoxication. Fortuitously, SORT1 is being explored as a target for ovarian cancer (Currie et al, 2022). Thus, improving the binding of $ACT_R$ to SORT1 through affinity maturation approaches may yield a novel solid tumor immunotoxin with all three native domains of ACT being used.

While this study was focused specifically on *pre-existing* ADAs to DT immunotoxins, reducing or eliminating pre-existing antitoxin antibodies is expected to additionally reduce the emergence of treatment associated ADA. Patients with pre-existing antitoxin antibodies, either from diphtheria vaccination (for DT immunotoxins) or *Pseudomonas* infection (for PE immunotoxins) develop anamnestic high-titer antibody responses after treatment with the associated immunotoxin (Wayne et al, 2014). Though ACT is not recognized by pre-existing ADA, it is likely that ADA to ACT would eventually develop during treatment owing to its bacterial origin. Various strategies have been explored for addressing treatment associated ADA. Recently, a directed mutagenesis study showed that B cell epitopes on DT could be reduced; seven mutations were made to DT that made it less immunogenic while retaining most of its activity (Schmohl et al, 2015). Notably, in ACT1 and ACT2, six of these seven residues are not conserved. More recent approaches to reduce treatment-associated ADA include co-dosing immunotoxins with tolerogenic nanoparticles containing rapamycin (Mazor et al, 2018) or low-dose methotrexate (King et al, 2018); these have shown success in inducing immune tolerance to immunotoxins. However, it was shown in the latter study that though methotrexate prevents memory recall response, it does not induce tolerance in mice with pre-existing antitoxin ADA (King et al, 2018).

Our discovery of ACT, a first-of-its-kind immunotoxin platform that retains the modularity and versatility of DT, is superior to DT in its ability to delivery heterologous cargos into cells; but, does not suffer from issues associated with pre-existing ADA, represents a promising new approach to usher in a new generation of immunotoxins with improved properties for cancer therapy.

# Methods

### Reagents and tools table

| Reagent/Resource | Reference or Source | Identifier or Catalog number |
|---|---|---|
| **Experimental models** | | |
| Vero-nLucP (Cercopithecus aethiops) | ATCC, Melnyk Lab (Park et al, 2018) | Cat # CCL-81 |
| Hap1 (H. sapiens) | Horizon Discovery | Cat # C631 |

| Reagent/Resource | Reference or Source | Identifier or Catalog number |
|---|---|---|
| HEK293T (H. sapiens) | ATCC | Cat # CRL-3216 |
| HPAF-II (H. sapiens) | ATCC | Cat # CRL-1997 |
| MV-4-11 (H. sapiens) | ATCC | Cat # CRL-9591 |
| SK-N-AS (H. sapiens) | ATCC | Cat # CRL-2137 |
| Single Donor Human Pediatric Serum age range 4–6 | Innovative Research | Cat # ISERSPED460 |
| **Recombinant DNA** | | |
| Champion™ pET SUMO Expression System | Thermo Fisher Scientific | Cat # K30001 |
| lentiCRISPRv2 | Addgene | Cat # 52961 |
| psPAX2 | Addgene | Cat # 12260 |
| pMD2.G | Addgene | Cat # 12259 |
| pLX301 | Addgene | Cat # 25895 |
| pET-28a | EMD Millipore Sigma | Cat # 69864 |
| **Antibodies** | | |
| Goat Anti-Human IgG+IgM+IgA H&L (HRP), 1/10,000 | Abcam | Cat # ab102420 |
| **Oligonucleotides and sequence-based reagents** | | |
| PCR primers | This study | Table EV1 |
| **Chemicals, enzymes and other reagents** | | |
| CloneAmp HiFi PCR Premix | Takara | Cat # 639298 |
| LB Broth Base (Lennox) | Invitrogen by Thermo Fisher Scientific | Cat # 12780052 |
| BL21(DE3) Competent E. coli | New England Biolabs | Cat # C2527H |
| NEB® 5-alpha Competent E. coli (High Efficiency) | New England Biolabs | Cat # C2987H |
| IPTG | Thermo Fisher Scientific | Cat # 34060 |
| protease inhibitor cocktail | Sigma Aldrich | Cat # P8849-5mL |
| Pierce™ universal nuclease inhibitor | Thermo Fisher Scientific | Cat # P88702 |
| Imidazole-HCl | Sigma Aldrich | Cat # I3386 |
| NaCl | Biotech | Cat # 7647-14-5 |
| Tris-HCl pH 8.0 | Wisent Inc. | Cat # 809-128-LL |
| SUMO Protease 1 | Life Sensor | Cat # SP-4010 |
| EMEM | Thermo Fisher Scientific | Cat # 11095080 |
| DMEM | Thermo Fisher Scientific | Cat # 11995073 |
| IMDM | Thermo Fisher Scientific | Cat # 12440053 |

| Reagent/Resource | Reference or Source | Identifier or Catalog number |
|---|---|---|
| MEM Non-Essential Amino Acids Solution (100X) | Thermo Fisher Scientific | Cat # 11140050 |
| FBS, Qualified | Thermo Fisher Scientific | Cat # 12483-020 |
| Penicillin-Streptomycin | Thermo Fisher Scientific | Cat # 15140122 |
| PrestoBlue Cell Viability Reagent | Thermo Fisher Scientific | Cat # A13262 |
| SYPRO Orange | Invitrogen by Thermo Fisher Scientific | Cat # S6651 |
| Tween®-20 | Sigma Aldrich | Cat # 1379-500mL |
| TBS 10X | Wisent Inc. | Cat # 311-030-LL |
| TMB Reagent | Life Technologies | Cat # 002023 |
| **Software** | | |
| GraphPad Prism 10.0.0 | GraphPad Software | https://www.graphpad.com |
| CFX Manager 3.1 software | BioRad | https://www.bio-rad.com/en-ca/sku/1845000-cfx-manager-software?ID=1845000 |
| BLOSUM62 | Henikoff and Henikoff (1992) | https://doi.org/10.1073/pnas.89.22.10915 |
| Phenix software package | Adams et al (2010) | https://www.phenix-online.org/ https://doi.org/10.1107/S0907444909052925 |
| Coot | Emsley et al (2010) | https://www2.mrc-lmb.cam.ac.uk/personal/pemsley/coot/ https://doi.org/10.1107/S0907444910007493 |
| Pymol | The PyMOL Molecular Graphics System, Version 1.8 Schrödinger, LLC. | https://pymol.org/2/#products |
| **Other** | | |
| SpectraMax M5e plate reader | Molecular Devices | n/a |
| Wizard® Genomic DNA Purification Kit | Promega | Cat # A1120 |
| CellTiter-Glo® 2.0 Cell Viability Assay | Promega | Cat # G9242 |
| CFX96 qRT-PCR thermocycler | BioRad | n/a |
| Nano-Glo® Luciferase Assay System | Promega | Cat # N1130 |
| Presto Mini Plasmid Kit | FroggaBio | Cat # PDH300 |
| Ni-NTA Magnetic Agarose | Thermo Fisher Scientific | Cat # 78606 |
| Nunc MaxiSorp™ plates | Thermo Fisher Scientific | Cat # 12565136 |
| 96-well assay plate | Corning | Cat # CLS3610 |

## Methods and protocols

### Cell culture and maintenance

All cell lines were maintained as per manufacturer instructions, in 37 °C incubators with 95% air and 5% $CO_2$. The green African monkey kidney cell line Vero-nLucP was maintained in DMEM (Thermo Fisher Scientific) with 10% fetal bovine serum (FBS, Thermo Fisher Scientific) and 1% penicillin-streptomycin (Thermo Fisher Scientific). Vero-nLucP cells were subcultured ever 3–4 days or at 80–90% confluency. Human near-haploid cells derived from the chronic myelogenous leukemia cell line KBM-7 (Hap1 cells), and human macrophage cells isolated from the blast cells of a biphenotypic B-myelomonocytic leukemia patient (MV-4-11) were both maintained in IMDM (Thermo Fisher Scientific) with 10% FBS and 1% penicillin-streptomycin. Hap1 cells were subcultured every 3–4 days, or at <70% confluency. MV-4-11 were maintained at <1,000,000 cells/mL. Human embryonic kidney cells (HEK392T) and human pancreatic adenocarcinoma (HPAF-II) cells were both maintained in EMEM (Thermo Fisher Scientific) with 10% FBS and 1% penicillin-streptomycin. They were both subcultured every 3–4 days or at 75–85% confluency. Of note, HEK293T were well resuspended before subculturing as they clump easily. Human neuroblasts (SK-N-AS) were maintained in DMEM (Thermo Fisher Scientific) with 10% FBS, 1% penicillin-streptomycin, and 0.1 mM non-essential amino acids (Thermo Fisher Scientific). SK-N-AS cells were subcultured every 5–7 days. All cell lines in this study were tested for mycoplasma 3 times per year and confirmed to not be contaminated.

### DT homolog screens—construct generation and purification

Accession numbers for all DT homologs in order of Fig. 2A, are as follows: WP_143115263.1/GAB79386.1, WP_116115734.1, WP_219106995.1, WP_120757473.1, SDT83331.1, WP_189053160.1, JZ58907.1, and WP_160159328.1. Of note, *A.chelonae* was originally separated into a DT$_A$-like and DT$_B$-component, by a one nucleotide insertion (accession number NZ_BAGZ01000024.1 G43277, compliment). This point insertion was removed prior to codon optimization.

Each sequence was aligned to the DT sequence to determine the identity percentage and boundary cut-offs for chimeric proteins shown in Fig. 2B,C. BLOSUM 62 was used to calculate percent identity. The DT$_C$ boundaries were from residues 1–185, the DT$_T$ boundaries were from residues 202–377, and the DT$_R$ boundaries were 378–535. Genes were cloned into the Champion™ pET SUMO *E. coli* expression system by Bon Opus Biosciences. To express and purify the proteins:

1. Plasmids obtained from Bon Opus Biosciences were transformed into BL21(DE3) cells (NEB) and plated on kanamycin LB-agar plates (50 μg/mL kanamycin). Plates were left in a 37 °C incubator overnight (16–20 h).
2. The next day, a colony was picked and inoculated into 5 mL of LB with 50 μg/mL kanamycin. This was incubated at 37 °C overnight (16–20 h), with shaking at 220 rpm.
3. The next day, 2 mL of the starter culture was inoculated into 35 mL of LB medium with 50 μg/mL kanamycin and induced with 1 mM IPTG at 25 °C for 4 h. The incubator was shaking at 220 rpm.
4. Cells were centrifuged at 5000 rpm and resuspended in 1 mL of lysis buffer (1% protease inhibitor cocktail, 0.01% Pierce™ universal nuclease inhibitor, 10 mM imidazole, 300 mM NaCl, 20 mM Tris-HCl pH 8.0).

5. Cells were lysed by sonication, with $3\times$ 10 s on 10 s off, at 10% amplitude.

6. Whole cell lysate was centrifuged at $18,000 \times g$.

7. The supernatant was incubated with Pierce™ Ni-NTA Magnetic Agarose for 45 min, at 4 °C and rotation.

8. The beads were collected using a magnetic rack, and washed $3\times$ with 300 mM NaCl, 10 mM Tris-HCl pH 8.0, 10 mM imidazole.

9. The protein was eluted with a 300 mM imidazole, 300 mM NaCl, Tris-HCl pH 8.0 buffer.

10. The protein was buffer exchanged into 300 mM NaCl, 10 mM Tris-HCl pH 8.0, and incubated with SUMO protease overnight at 4 °C, to cleave the 6xHis-SUMO affinity tag.

11. The protein was incubated with Pierce™ Ni-NTA Magnetic Agarose to remove SUMO protease and the SUMO tag, and the flowthrough (protein) was collected.

12. The protein concentration was estimated by $A_{280}$ using each recombinant proteins respective molecular weight and extinction coefficient.

### DT homolog screens—protein synthesis assay

1. Vero-nLucP cells that had been engineered as previously described (Park et al, 2018; Sugiman-Marangos et al, 2022b) were plated at 5000 cells/well in 96-well white clear bottom plates (Corning).

2. The following day, protein toxin was serially diluted at 10X the experiment concentration, in cDMEM (DMEM + 10% FBS + 1% penicillin-streptomycin) and was added to cells at 1X (spiking in 10 μL into experiment plate that has 90 μL media).

3. Cell were incubated for 24 h at 37 °C.

4. The next day, cells were read for luminescence signal using the NanoGlo® Luciferase Assay kit (Promega), on a SpectraMax M5e plate reader (Molecular Devices). Data was corrected to untreated cells (100% nanoluciferase signal) by dividing the reading from the treated cells by the untreated cells, and multiplying by 100. $EC_{50}$ values were determined by Prism software (GraphPad).

### Crystallization and structure determination of ACT1

1. The ACT1 gene is from accession numbers WP_143115263.1 and GAB79386.1.

2. The E. coli codon optimized gBlock gene fragment was ordered from Integrated DNA Technologies with overhangs to the Champion™ pET SUMO expression vector, and cloned into the Champion™ pET SUMO E. coli expression system by Gibson Assembly (Table EV1).

3. The ACT1 plasmid was transformed into BL21(DE3) cells (NEB) and plated on kanamycin LB agar plates (50 μg/mL kanamycin) and incubated overnight in a 37 °C incubator, for 16–20 h.

4. The next day, a colony was inoculated into a 50 mL starter culture (LB+kanamycin) and incubated overnight at 37 °C, shaking at 220 rpm, for 16–20 h.

5. The next day, the overnight culture was inoculated into 1 L of LB medium + kanamycin and induced with 0.1 mM IPTG at 18 °C for 18 h.

6. Cells were centrifuged at 5000 rpm and resuspended in lysis buffer (1% protease inhibitor cocktail, 1 mg/mL lysozyme, 0.01% Pierce™ universal nuclease inhibitor, 20 mM imidazole, 500 mM NaCl, 20 mM Tris-HCl pH 7.5).

7. Cells were lysed with three passes through an Emulsiflex C3 (Avestin) at 15,000 psi.

8. Whole cell lysate was centrifuged at $18,000 \times g$ and the supernatant was filtered through a 0.45 μm filter and passed over a HisTrap FF crude column (Cytiva).

9. The protein was eluted with 50–75 mM imidazole, buffer exchanged into 150 mM NaCl, 20 mM Tris-HCl pH 7.5, and incubated with SUMO protease overnight at 4 °C, to cleave the 6xHis-SUMO affinity tag.

10. The protein was flowed over a HisTrap FF crude column and the flowthrough (protein) was collected and concentrated to 8 mg/mL by centrifugation.

11. The protein concentration was estimated by $A_{280}$ using ACT1s molecular weight and extinction coefficient.

12. Hanging drop vapor diffusion was used to grow crystals. The condition in which ACT1 crystals were obtained contained 2 μL of mother liquor (0.2 M calcium chloride, 0.1 M Tris-HCl pH 8.5, 25% (w/v) PEG4000) and 1 μL of 8 mg/mL protein.

13. The drop was dehydrated over 130 μL of 2 M $(NH_4)_2PO_4$ for 45 min prior to freezing in liquid nitrogen.

14. Data was collected at the Advanced Photon Source on the 23-ID-D beamline at a wavelength of 1.03319 Å.

15. Initial phases were determined using Phaser in the Phenix software package by using a multi-component search models with individual DT domains (C-domain residues 13–167, R-domain residues 391–531, T-domain residues 205–378) in which disordered loops had been removed. The structure was refined using iterative cycles of phenix.refine and autobuild.

16. Data collection and model refinement statistics are in Appendix Table S1.

### Structure determination of ACT2 using AlphaFold 2.0

The google.colab notebook was used to obtain the structure of ACT2 (Jumper et al, 2021; Mirdita et al, 2022). Sequence from accession number WP_116115734.1 was used as the query sequence, and the pdb100 was set as the template mode.

### Receptor-binding domain cell sensitivity assays

1. Receptor-binding domain boundaries for ACT1 and ACT2 were determined by aligning each sequence to $DT_R$.

2. Cloning was performed by Bon Opus Biosciences into the Champion™ pET SUMO E. coli expression system, and proteins were expressed similar to the DT homolog screen.

3. Hap 1 cells were seeded in a 96-well plate one day prior to application of toxin at ~35–40% confluency.

4. Serial dilutions of individual toxins were made in storage buffer (150 mM NaCl, 20 mM Tris pH 7.5, 5% glycerol) at 10X the experiment concentration and incubated with cells at 1X, for 48 h.

5. Toxin sensitivity was measured with CellTiter-Glo reagent (Promega) according to manufacturer protocol.

6. Viability was normalized to untreated wells prior to data visualization with Prism software (GraphPad).

### Virus generation

1. To generate lentivirus for the construction of the Hap1 TKOv3 library (Hart et al, 2017), a ~1:1:1 molar ratio mixture of library transfer (lentiCRISPRv2; Addgene plasmid #52961) and packaging plasmids (psPAX2, Addgene #12260; pMD2.G, Addgene #12259) was prepared in serum-free media (Opti-MEM™; Gibco™, cat # 31985062).

2. A ~3:1 ratio of X-tremeGENE™ 9 DNA Transfection Reagent (Roche, XTG9-RO) was added to the mixture prior to incubation and application on HEK293T cells using standard methods (Hart et al, 2017).

3. Virus was collected 48 h after infection.

4. To generate lentivirus for single gene knockout lines, gRNAs were selected from the TKOv3 library and cloned into lentiCRISPRv2 under established protocol (Hart et al, 2017).

5. Packaging and lentiCRISPRv2 plasmids were transfected into HEK293Ts using X-tremeGENE™ 9 as above.

6. Similarly, to generate lentivirus for HEK293T SORT1 over-expression cell lines, SORT1 cDNA was obtained from the hORFeome V8.1 collection and cloned into pLX301 (Addgene #25895) for co-transfection with psPAX2 and pVSV-G (Addgene #138479).

7. Virus was collected from HEK293Ts 72 h post-transfection for all gene-specific constructs.

## Hap1 genome wide CRISPR KO screens

1. To generate a starting (T0) cell population with ~200-fold gRNA library coverage, 50 million Hap1s were seeded and transduced the same day with the lentiviral TKOv3 library at MOI 0.3 under 8 µg/mL polybrene.

2. The day after infection, fresh media containing 1 µg/mL puromycin was applied for 48 h to eliminate cells without gRNA inserts.

3. Surviving T0 cells were pooled, and the library was maintained at minimum ~100-fold coverage beyond this point.

4. Cells underwent expansion before 7 million cells were seeded into two 15 cm plates per condition on T4.

5. Toxin was applied to cells the next day (T5); at T7, toxin-containing media was removed and replaced with complete growth media (IMDM, 10% FBS) to allow survivor repopulation.

6. Untreated cells were passaged every 72 h in parallel with the replacement of fresh media on toxin-treated plates.

7. Survivors were reseeded once colonies were visible and collected when plates reached ~80% confluency alongside untreated cells from the same day.

## Next-generation sequencing library preparation

Genomic DNA was extracted using the Wizard® Genomic DNA Purification Kit (Promega, cat #A1120) following manufacturer protocol. gRNA inserts were then amplified from each sample. Amplicons were barcoded with Illumina TruSeq adapters i5 and i7 sequences prior to NGS sequencing at a read depth of at least 5 million reads per sample. Hits were identified from FASTQ files and ranked using MAGeCK software (Li et al, 2014).

## Generation of stable cell lines for screen validation

1. Hap1 or HEK293Ts were seeded in 6-well plates at <40% confluency.

2. The next day, cells were transduced with construct-specific virus and 8 µg/mL polybrene.

3. After overnight incubation, virus-containing media was removed, and cells were selected with 2 µg/mL puromycin for 48 h.

4. Each polyclonal stable cell population was reseeded in 10 cm plates for expansion.

5. Monoclonal cell lines were obtained from polyclonal populations through serial dilutions in 96-well plates.

6. Clones were validated through immunoblot for target proteins.

## ELISA with human sera

1. Nunc MaxiSorp™ plates (Thermo Fisher Scientific) were immobilized with 1 µg of protein overnight at 4 °C, in carbonate pH 9.4 buffer.

2. They were then blocked with 3% milk in TBS (TBSM) overnight at 4 °C.

3. Dilutions of human serum (Human Serum age 4–6, Innovative Research) were performed in TBSM, and wells were incubated with 100 µL for 2 h at 37 °C. Of note, various human patient samples were used in the ELISAs, and immunization status of the patients was unknown.

4. Wells were then washed with TBST (0.1% tween 20) and then incubated with an anti-human IgG antibody conjugated to HRP (Abcam, ab102420) for 1 h at 37 °C.

5. Wells were developed using 100 µL of TMB reagent (Thermo Fisher Scientific).

6. Absorbance was read at 630 nm.

7. Background was twice the A630 reading of control wells (no-protein, +human serum; bovine serum albumin, + human serum).

8. The titer at which the experiment line crossed background was plotted.

## Serum toxicity assays

1. Protein toxins were incubated with either human sera (Human Serum age 4–6, Innovative Research) or PBS in a 1:1 ratio, for 30 min at room temperature.

2. Sample was then added to cells that had been plated at 10,000 cells/well the previous day, in a 96-well white clear bottom plate (Corning).

3. Cells were incubated for 72 h at 37 °C, upon which cells were read for viability.

4. Viability of MV-4-11 cells was assessed using the CellTiterGlo 2.0 kit from Promega (as per manufacturer instructions); viability of HPAFII cells was assessed using PrestoBlue reagent from Thermo Fisher (as per manufacturer instructions).

5. Values were corrected to sera-only treated cells, which represented 100% viability.

## CD123, Her3-integrin αvβ6, and TEM8 targeting immunotoxin generation

1. The immunotoxins targeting CD123 use an scFv as the receptor binding domain. The scFv was derived from the antibody used in the ADC IMGN632 (Kovtun et al, 2018)—the variable light chain ($V_L$) was linked to the variable heavy chain ($V_H$) via a $(G_4S)_3$ linker. The $V_L$-$V_H$ scFv was linked to the bacterial toxin also via a $(G_4S)_3$ linker.

2. DT-CD123 was made using the first 389 residues of DT.

3. ACT-CD123 was made using residues 32–217 from ACT-TVS, 186–201 from DT, and 204–379 from ACT2. Cloning was performed by Bon Opus Biosciences, into the pET SUMO *E. coli* expression system.

4. The scFv targeting TEM8 was derived from the m825 antibody described in Szot et al similar to the CD123, except the orientation was $V_H$-$V_L$.

5. The immunotoxin targeting Her3- integrin αvβ6 was synthesized using residues 1–186 from ACT1, 186–201 from DT, and 204–379 from ACT2.

6. The Her3 targeting affibody was from Nazari et al (2019) and the integrin αvβ6 peptide was from DiCara et al (2007). All cloning

**The paper explained**

**Problem**

Nearly all patients receiving diphtheria toxin (DT)-based immunotoxin therapies have pre-existing anti-drug antibodies (ADAs) due to childhood vaccinations against diphtheria toxin. These pre-existing ADAs bind and neutralize DT-based immunotoxins, resulting in inconsistent and unpredictable exposure, efficacy, and safety of these potent cancer drugs. Novel strategies are urgently needed to circumvent pre-existing ADAs to broaden the use of DT-based immunotoxins for treating liquid and solid tumors.

**Results**

In this study, we describe the discovery of a distant homolog of DT from the reptile pathogen *Austwickia chelonae*, termed chelona toxin (ACT). We show that ACT has structural and functional similarities to DT but is not recognized by pre-existing anti-DT antibodies in human sera. Like DT, ACT can successfully be re-targeted to tumor-associated receptors creating novel ACT-immunotoxins that—unlike DT-immunotoxins—are not neutralized by human serum. Moreover, we found that ACT's translocase domain was superior to the DT translocase in delivering various alternative therapeutic protein cargos into cells by more efficiently escaping endosomes after internalization.

**Impact**

These findings mark a significant step in the immunotoxin field, addressing a major liability that has limited the effectiveness and broad adoption of current FDA approved immunotoxins. Replacing DT with ACT in future immunotoxin design could improve cancer patient outcomes and reinvigorate efforts to develop potentially a new arsenal of treatment options for various cancers, including solid tumors.

was performed by Bon Opus Biosciences, into the pET SUMO *E. coli* expression system.

7. Proteins were expressed and purified similar to the DT homolog screen. TEM8 targeting constructs were expressed at 18 °C for 18 h.

### Generation of constructs delivering various cargo

To generate the constructs depicted in Fig. 4C, the $DT_C$, $DT_T$, $ACT1_T$, and $ACT2_T$ boundaries were chosen as described above, through sequence alignments. The Her3 and Tem8 targeting affibody and scFv (respectively) are the same as described above. The sequence for TcdB-GTD is residues 1–543 from accession number VFG96748.1. The sequence for TpeL-GTD is residues 1–543 from accession number BAF46125.1. Cloning was done by Bon Opus Biosciences, and proteins were purified as described in the DT homologs screen section.

### Differential scanning fluorimetry (DSF)

DSF with DT and ACT1 was performed as previously described (Sugiman-Marangos et al, 2022b). Briefly,

1. DT or ACT1 proteins were diluted in 35 µL of 150 mM citrate phosphate buffer containing 5X SYPRO Orange (Invitrogen), to a final concentration of 0.1 mg/mL DT or 0.4 mg/mL ACT1.
2. The pH of the buffer varied from 4.0–7.5, in 0.5 pH increments.
3. Fluorescence of SYPRO Orange was measured using a BioRad CFX96 qRT-PCR thermocycler, and the BioRad CFX Manager 3.1 software was used to determine melting temperatures.
4. Melting temperature data was normalized to the melting temperature at pH 7.5 for each respective protein.

### Kinetics assay

1. HCT116 cells were plated at 10,000 cells/well, in a 96-well white clear bottom plate (Corning).
2. The next day, 33 nM of each protein toxin (purified the same as the DT homologs screen) was incubated on cells for either 1, 3, 6, or 12 h, after which the media was changed for toxin-free media and left until 72 h.
3. ATP levels were measured using CellTiter-Glo reagent (Promega).
4. Values were corrected to media-only treated cells, which represented 100% viability.

### Crystal structure of TpeL-GTD

1. The TpeL-GTD gene was cloned from accession number BAF46125.1 (residues 1–543), into the pET-28a expression plasmid (EMD Millipore).
2. Protein was expressed and purified as described above for ACT1 expression and purification. The 6xHis tag was not removed.
3. Hanging drop vapor diffusion was used to grow crystals. TpeL-GTD crystallized in 0.2 M potassium sodium tartrate and 20% (w/v) PEG 3350 (1 µL mother liquor, 1 µL of 16 mg/mL protein).
4. Crystals were crushed and re-seeding into the mother liquor for optimization.
5. Crystals were frozen in liquid nitrogen.
6. Data was collected at the Advanced Photon Source on the 23-ID-D beamline at a wavelength of 1.0332 Å.
7. Initial phases were determined using Phaser in the Phenix software package by molecular replacement with the structure from PDB ID 4DMV (the GTD from *C. difficile* toxin A).
8. The structure was refined using iterative cycles of phenix.refine and autobuild, to a final resolution of 1.9 Å. Data collection and model refinement statistics are in Appendix Table S1.

# Data availability

All structural data generated during the current study for ACT1 and TpeL-GTD can be found in the Protein Data Bank (https://www.rcsb.org) under the accession numbers 9BIW and 9BON.

The source data of this paper are collected in the following database record: biostudies:S-SCDT-10_1038-S44321-024-00116-z.

# Peer review information

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

## Acknowledgements

The authors thank members of the Melnyk and Taipale labs for critical comments on this manuscript. We wish to thank the Structural and Biophysical Core at the Hospital for Sick Children. This work was supported by grants from the Canadian Institutes of Health Research (CIHR) (to RAM and MT) (Application number: 452580), and the Natural Sciences and Engineering Research Council of Canada RGPIN-2023-05371 (to RAM).

## Author contributions

**Shivneet K Gill**: Conceptualization; Data curation; Formal analysis; Investigation; Methodology; Writing—original draft; Writing—review and editing. **Seiji N Sugiman-Marangos**: Conceptualization; Data curation; Formal analysis; Investigation; Methodology. **Greg L Beilhartz**: Conceptualization; Formal analysis; Investigation; Methodology. **Elizabeth Mei**: Data curation; Formal analysis; Investigation. **Mikko Taipale**: Supervision; Funding acquisition. **Roman A Melnyk**: Conceptualization; Formal analysis; Supervision; Funding acquisition; Investigation; Visualization; Methodology; Writing—original draft; Project administration; Writing—review and editing.

Source data underlying figure panels in this paper may have individual authorship assigned. Where available, figure panel/source data authorship is listed in the following database record: biostudies:S-SCDT-10_1038-S44321-024-00116-z.

## Disclosure and competing interests statement

A patent on 'therapeutic applications of DT homologs' was filed by RAM, GLB, SNS-M, and SKG. RAM is a co-founder of Nighthawk Therapeutics.

