## [Peer Review File · EMBO Molecular Medicine]

An enhanced intracellular delivery platform based on a distant diphtheria toxin homolog that evades pre-existing anti-toxin antibodies

Shivneet Gill, Seiji Sugiman-Marangos, Greg Beilhartz, Elizabeth Mei, Mikko Taipale, and Roman Melnyk

Corresponding author: Roman Melnyk (roman.melnyk@sickkids.ca)

Review Timeline:

Submission Date:	2nd May 24
Editorial Decision:	22nd May 24
Revision Received:	2nd Jul 24
Editorial Decision:	5th Jul 24
Revision Received:	15th Jul 24
Accepted:	18th Jul 24

Editor: Lise Roth

Transaction Report:

22nd May 2024

Dear Prof. Melnyk,

Thank you for the submission of your manuscript to EMBO Molecular Medicine. We have now received feedback from the three reviewers who agreed to evaluate your manuscript. As you will see from the reports below, the referees acknowledge the interest of the study and are overall supporting publication of your work pending appropriate revisions.

Addressing the reviewers' concerns in full will be necessary for further considering the manuscript in our journal, and acceptance of the manuscript will entail a second round of review.

Please note that upon further cross-commenting, the referees agreed that the last point mentioned by referee #2 ("potential concern of anti-CT Abs elicited during the therapy, which may reduce the effect of the treatment. Can this be tested experimentally using mouse sera enriched for anti-CT antibodies?") could be addressed by adequate discussion, and not necessarily experimentally.

EMBO Molecular Medicine encourages a single round of revision only and therefore, acceptance or rejection of the manuscript will depend on the completeness of your responses included in the next, final version of the manuscript. For this reason, and to save you from any frustrations in the end, I would strongly advise against returning an incomplete revision.

We are expecting your revised manuscript within three months, if you anticipate any delay, please contact us.

We require:

4) A .docx formatted letter INCLUDING the reviewers' reports and your detailed point-by-point responses to their comments. As part of the EMBO Press transparent editorial process, the point-by-point response is part of the Review Process File (RPF), which will be published alongside your paper.

5) A complete author checklist, which you can download from our author guidelines (<https://www.embopress.org/page/journal/17574684/authorguide#submissionofrevisions>). Please insert information in the checklist that is also reflected in the manuscript. The completed author checklist will also be part of the RPF.

6) It is mandatory to include a 'Data Availability' section after the Materials and Methods. Before submitting your revision, primary datasets produced in this study need to be deposited in an appropriate public database, and the accession numbers and database listed under 'Data Availability'. Please remember to provide a reviewer password if the datasets are not yet public (see <https://www.embopress.org/page/journal/17574684/authorguide#dataavailability>).

7) For data quantification: please specify the name of the statistical test used to generate error bars and P values, the number (n) of independent experiments (specify technical or biological replicates) underlying each data point and the test used to calculate p-values in each figure legend. The figure legends should contain a basic description of n, P and the test applied. Graphs must include a description of the bars and the error bars (s.d., s.e.m.). Please provide exact p values.

8) Our journal encourages inclusion of *data citations in the reference list* to directly cite datasets that were re-used and obtained from public databases. Data citations in the article text are distinct from normal bibliographical citations and should

directly link to the database records from which the data can be accessed. In the main text, data citations are formatted as follows: "Data ref: Smith et al, 2001" or "Data ref: NCBI Sequence Read Archive PRJNA342805, 2017". In the Reference list, data citations must be labeled with "[DATASET]". A data reference must provide the database name, accession number/identifiers and a resolvable link to the landing page from which the data can be accessed at the end of the reference. Further instructions are available at .

9) We replaced Supplementary Information with Expanded View (EV) Figures and Tables that are collapsible/expandable online. A maximum of 5 EV Figures can be typeset. EV Figures should be cited as 'Figure EV1, Figure EV2" etc... in the text and their respective legends should be included in the main text after the legends of regular figures.

10) The paper explained: EMBO Molecular Medicine articles are accompanied by a summary of the articles to emphasize the major findings in the paper and their medical implications for the non-specialist reader. Please provide a draft summary of your article highlighting

11) For more information: There is space at the end of each article to list relevant web links for further consultation by our readers. Could you identify some relevant ones and provide such information as well? Some examples are patient associations, relevant databases, OMIM/proteins/genes links, author's websites, etc...

12) Author contributions: CRediT has replaced the traditional author contributions section because it offers a systematic machine readable author contributions format that allows for more effective research assessment. Please remove the Authors Contributions from the manuscript and use the free text boxes beneath each contributing author's name in our system to add specific details on the author's contribution. More information is available in our guide to authors.

13) Disclosure statement and competing interests: We updated our journal's competing interests policy in January 2022 and request authors to consider both actual and perceived competing interests. Please review the policy <https://www.embopress.org/competing-interests> and update your competing interests if necessary.

14) Every published paper now includes a 'Synopsis' to further enhance discoverability. Synopses are displayed on the journal webpage and are freely accessible to all readers. They include a short stand first (maximum of 300 characters, including space) as well as 2-5 one-sentences bullet points that summarizes the paper. Please write the bullet points to summarize the key NEW findings. They should be designed to be complementary to the abstract - i.e. not repeat the same text. We encourage inclusion of key acronyms and quantitative information (maximum of 30 words / bullet point). Please use the passive voice. Please attach these in a separate file or send them by email, we will incorporate them accordingly.

15) As part of the EMBO Publications transparent editorial process initiative (see our Editorial at <http://embomolmed.embopress.org/content/2/9/329>), EMBO Molecular Medicine will publish online a Review Process File (RPF) to accompany accepted manuscripts.

In the event of acceptance, this file will be published in conjunction with your paper and will include the anonymous referee reports, your point-by-point response and all pertinent correspondence relating to the manuscript. Let us know whether you agree with the publication of the RPF and as here, if you want to remove or not any figures from it prior to publication. Please note that the Authors checklist will be published at the end of the RPF.

I look forward to receiving your revised manuscript.

Yours sincerely,

Lise Roth

***** Reviewer's comments *****

Referee #1 (Remarks for Author):

The manuscript by Gill et al. presents an interesting study about engineering a successor immunotoxin to the established agents Ontak, Elzonris or Lumoxiti. These have the drawback that DT toxoid vaccinated people developed antibodies which limit the success of the immunotoxins. By swapping domains between diphtheria toxin and related toxins of *Austwickia chelonae* the authors created similar immunotoxins avoiding this drawback. This interesting study is technically sound and clearly written. My only criticism:

It would be nice to have information about the toxoid vaccine used. Moreover, how fast is antibody-development in non-vaccinated people?

Line39: What are cancer receptors? Did you mean receptors highly expressed on cancer cells?

Line 40/41: What is the advantage using protein toxins compared to toxic small molecules?

Lines 136 and following: Why is it necessary to know the receptor for CT1 and CT2? The receptor domain is replaced anyway. Would it better to use "cargos" instead of "cargo" at many positions of the manuscript?

Statistics are missing!

Referee #2 (Remarks for Author):

In this paper the authors report the discovery and characterization of distant relatives of diphtheria toxin, named chelona toxins (CT1 and CT2), which are produced by an ancient reptile pathogen. Interestingly -despite the close similarity to DT- these molecules are not affected by the anti-diphtheria antibodies which are present in most of the population and can be used to deliver the toxic subunit of diphtheria toxin to target cancer cells in a way that is even more efficient than DT-derived immunotoxins. Indeed, the presence of anti-diphtheria toxin antibodies which are induced by diphtheria vaccination is a major obstacle to the use of DT-based immunotoxins as anti cancer interventions.

The study is well done and the results are extensive, ranging from primary sequence analysis, 3D structure determination and biological data generated by means of ad-hoc constructs, aimed at dissecting the functionality of each of the 3 domains of the novel toxins.

This work has potentially a great impact as it proposes a solution that is likely to improve the efficacy of anti-cancer drugs based on immunotoxins.

Major comments

- The authors abbreviate the new toxin as CT. I strongly encourage the authors to change the abbreviation as "CT" is used in the literature to indicate cholera toxin which is an ADP-ribosylating toxin of the same family of diphtheria toxin and may therefore be confusing to the reader.
- Diphtheria toxin is indeed a versatile molecule used for different purpose in treatment and prevention. One of the most widely adopted use of DT is as carrier molecule for glycoconjugated vaccines. Indeed, different inactivated forms of DT have been used for the development of multi-valent meningococcal and pneumococcal vaccines, now licensed and widely used throughout the world, especially in infants and young children. The anti-DT antibodies present in the human population have also been reported as potential cause of a diminished immunogenicity to DT-conjugated polysaccharides antigens as the valency of novel vaccines increases (the so-called carrier suppression effect mechanism). For this reason, novel carrier molecules are being actively investigated. Although this is a completely different topic, it would still be valuable and interesting if the authors could comment in the Discussion about the assessment of these DT-like molecules as novel potential carriers for conjugated vaccines.
- The authors initially mention two potential candidate toxins, CT1 and CT2, but eventually the final experiments have been performed with constructs indicating CT. Can you please better indicate the process for final candidate selection? Is it CT1 or

CT2? And why?

- In the Discussion, the authors mention the potential concern of anti-CT Abs elicited during the therapy, which may reduce the effect of the treatment. Can this be tested experimentally using mouse sera enriched for anti-CT antibodies?

Minor Comments:

- In Fig 2a, please provide % seq id for every domain (C, T and R)
- In Fig 2b, 2c: why first graph has 8 chimera candidates, and second graph has 7 candidates? What is the benchmark value of EC50 by wt DT?
- Line 92: the word "time" is missing
- Lines 208-210: please rephrase as sentence does not read well
- Line 242, the word "antibodies " is missing
- Lines 255-257: please rephrase sentenc

Referee #3 (Remarks for Author):

This work present an alternative to the diphtheria toxin (DT) for the construction of immunotoxins that would escape preexisting antibodies from vaccination. DT led to FDA approved immunotoxins but anti-drug antibodies (ADA) remain a challenge. The authors search data banks for DT analogues and showed that toxins (chelonatoxins) from a reptile pathogen could be alternatives to DT. This was demonstrated by retargeting these toxins to alternate receptors expressed by AML or tumor cells. In addition, the authors show that the T domain from chelonatoxin can translocate catalytic domain from other toxins more efficiently than DT. Although interesting, the importance of this part seems relative: why change a catalytic toxic domain from the native toxin to that of another toxin affecting small G proteins ? This part should be better explained in the discussion in terms of applicability in biomedicine. Previous experiences showed that most proteins cannot accept the translocation process of DT, limiting applicability of this principle. The authors should also better discuss the comparative potency of the new immunotoxins with respect to the classical DT, which is very powerful in terms of cell toxicity.

Minor: please correct typo line 184

Point-by-point response to referee comments

Dear Dr. Roth,

Below, please see our point-by-point responses to each of the referee comments (in red).

***** Reviewer's comments *****

Referee #1 (Remarks for Author):

The manuscript by Gill et al. presents an interesting study about engineering a successor immunotoxin to the established agents Ontak, Elzonris or Lumoxiti. These have the drawback that DT toxoid vaccinated people developed antibodies which limit the success of the immunotoxins. By swapping domains between diphtheria toxin and related toxins of *Austwickia chelonae* the authors created similar immunotoxins avoiding this drawback. This interesting study is technically sound and clearly written.

My only criticism:

It would be nice to have information about the toxoid vaccine used.

In the current study, we used human sera that were collected from individuals that were between the ages of 4-6 at FDA-licensed collection centers across the United States. Given the age of the individuals these individuals likely received the DTaP vaccine per the standard US vaccination protocols; however, because these were purchased from Innovative Research and there was no patient information attached, we do not know the exact vaccination status of these patients. For instance, despite being collected in the US, some patients may have been vaccinated in a different country using a slightly different vaccine. To mitigate this, we tested sera from both individuals and from pooled samples.

...Moreover, how fast is antibody-development in non-vaccinated people?

This is a great question and one that we searched extensively to find, however, we were unsuccessful in finding this information. This is due in part because there are so few unvaccinated individuals in the developed world.

Line39: What are cancer receptors? Did you mean receptors highly expressed on cancer cells?

Yes – thank you for catching this. We have modified the text on Line 39/40.

Line 40/41: What is the advantage using protein toxins compared to toxic small molecules?

The most notable advantage of protein toxins compared to toxic small molecules, is that the cytotoxic payloads of immunotoxins cannot go on to intoxicate neighboring (healthy) cells (aka the “bystander effect”). Once the cytotoxic catalytic domain is released into the cytosol, this enzyme cannot, on its own, enter neighboring cells without its associated translocation moiety. Toxic small molecules on the other hand, can more easily diffuse into neighboring cells after the target cell dies, which often contributes to

their off-target toxicity. Another advantage of protein toxins over small molecule toxins is greater potency owing to the catalytic mechanism by which protein toxins inactivate their intracellular targets. On the other hand, small molecule toxins have certain benefits over protein toxins in terms of cost of goods and potentially immunogenicity. There are many more distinctions between each modality that make one or the other advantageous for the particular indication.

Rather than focus on advantages and potential disadvantages of immunotoxins vs ADCs in the introduction, a sentence was added in line 41 that highlights the distinction between the two modalities with respect to the bystander effect. A reference to a review discussing the differences in ADCs and immunotoxins by Pastan and co-workers was also added.

Lines 136 and following: Why is it necessary to know the receptor for CT1 and CT2? The receptor domain is replaced anyway.

We think that it is worth including data related to elucidating the native receptor-binding moieties of the Chelona toxins in part to provide insights into the host range of these toxins and indirectly the evolution of these toxins. This is interesting both from both a basic fundamental understanding of the molecular evolution of this family of toxins, and also from a practical perspective to understand if the lack of cross-reactive immunity to CT1 and CT2 by anti-DT ADA could be a potential problem clinically. For instance, if a human were to become infected by a toxigenic strain of *Austwickia cheloniae*, knowing that we are not protected from its toxins, which do bind a human receptor, albeit weakly, could be important information for devising antitoxin approaches to prevent disease (i.e., using receptor decoys). Moreover, showing that these toxins bind the human SORT1 receptor may also open up additional avenues of research by others or us to characterize the binding of these toxins to a related receptor in a different host species that is yet to be determined. We acknowledge that this is not a major part of the paper, which we feel is reflected by the relatively limited amount of text and figures devoted to this aspect.

Would it better to use "cargos" instead of "cargo" at many positions of the manuscript? Statistics are missing!

Thank you for catching this. We have changed this in the manuscript every time it appeared as appropriate. The statistics are included in the figure legends.

Referee #2 (Remarks for Author):

In this paper the authors report the discovery and characterization of distant relatives of diphtheria toxin, named chelona toxins (CT1 and CT2), which are produced by an ancient reptile pathogen. Interestingly -despite the close similarity to DT- these molecules are not affected by the anti-diphtheria antibodies which are present in most of the population and can be used to deliver the toxic subunit of diphtheria toxin to target cancer cells in a way that is even more efficient than DT-derived immunotoxins. Indeed, the presence of anti-diphtheria toxin antibodies which are induced by diphtheria vaccination is a major obstacle to the use of DT-based immunotoxins as anti cancer

interventions.

The study is well done and the results are extensive, ranging from primary sequence analysis, 3D structure determination and biological data generated by means of ad-hoc constructs, aimed at dissecting the functionality of each of the 3 domains of the novel toxins.

This work has potentially a great impact as it proposes a solution that is likely to improve the efficacy of anti-cancer drugs based on immunotoxins.

Major comments

- The authors abbreviate the new toxin as CT. I strongly encourage the authors to change the abbreviation as "CT" is used in the literature to indicate cholera toxin which is an ADP-ribosylating toxin of the same family of diphtheria toxin and may therefore be confusing to the reader.

This is a fair point. To address this we considered how to best abbreviate "Chelona Toxin" in a way that was unique, clear, specific, simple and future proofed. It proved to be more difficult than we anticipated! A possibility we considered initially was to use a similar strategy as the large clostridial toxin field of using the formalism "Toxin-genus-species-identifier". In this regard, Chelona Toxin 1 and Chelona Toxin 2 would be Tac1 and Tac2, respectively. This is unique and specific and future proofed, however Tac1 is both the name of the gene that encodes Preprotachykinin precursor 1 and another gene in Arabidopsis. The simplest possibility that we landed on was using "ACT" for Austwickia "Chelona Toxin" – and use ACT1 and ACT2 for the two chelona toxins we most prominently characterized. We have made changes throughout the manuscript from CT to ACT. We have also changed the nomenclature in the figures (Figs 2-5).

- Diphtheria toxin is indeed a versatile molecule used for different purpose in treatment and prevention. One of the most widely adopted use of DT is as carrier molecule for glycoconjugated vaccines. Indeed, different inactivated forms of DT have been used for the development of multi-valent meningococcal and pneumococcal vaccines, now licensed and widely used throughout the world, especially in infants and young children. The anti-DT antibodies present in the human population have also been reported as potential cause of a diminished immunogenicity to DT-conjugated polysaccharides antigens as the valency of novel vaccines increases (the so-called carrier suppression effect mechanism). For this reason, novel carrier molecules are being actively investigated. Although this is a completely different topic, it would still be valuable and interesting if the authors could comment in the Discussion about the assessment of these DT-like molecules as novel potential carriers for conjugated vaccines.

This is interesting. Of course, a potential issue with using a toxin like Chelona Toxin as a novel carrier molecule is that this would be directly at odds with what we are trying to accomplish with using CT as a novel immunotoxin scaffold to evade pre-existing ADA arising from prior vaccination. In other words, this would solve one problem potentially, but create another. Also, at this time, we have no way of knowing if ADA against other non-CT carriers would be cross-reactive with CT and thereby make CT prone to pre-existing ADA. We feel that more data is required to support this and is outside of the scope of the current work.

- The authors initially mention two potential candidate toxins, CT1 and CT2, but eventually the final experiments have been performed with constructs indicating CT. Can you please better indicate the process for final candidate selection? Is it CT1 or CT2? And why?

We wish to thank this referee for catching this. During the development of this story, we found that CT1 and CT2 were virtually interchangeable in all contexts and as such, we inadvertently referred to both as CT. However, we realize this is not rigorous and have gone back to indicate in the figures and figure legends where ACT1 or ACT2 and/or domains thereof were used.

- In the Discussion, the authors mention the potential concern of anti-CT Abs elicited during the therapy, which may reduce the effect of the treatment. Can this be tested experimentally using mouse sera enriched for anti-CT antibodies?

We understand that the Referees and Editor have discussed this particular point during the review process. We have dealt with this topic in the Discussion starting on line 274.

Minor Comments:

- In Fig 2a, please provide % seq id for every domain (C, T and R)

% seq ID's have been inserted into Figure 2a

- In Fig 2b, 2c: why first graph has 8 chimera candidates, and second graph has 7 candidates? What is the benchmark value of EC50 by wt DT?

We were unable to express one of the T-domain chimeras – and therefore could not assess it's activity as indicated. The benchmark EC50 values for wt DT in 2b and 2c are 0.6 ± 0.2 pM and 0.70 ± 0.2 pM. We have these values into the figure legend for Figure 2 as a reference.

- Line 92: the word "time" is missing

Done. Thanks!

- Lines 208-210: please rephrase as sentence does not read well

Done. Thanks!

- Line 242, the word "antibodies " is missing

Done. Thanks!

- Lines 255-257: please rephrase sentenc

Done. Thanks!

Referee #3 (Remarks for Author):

This work present an alternative to the diphtheria toxin (DT) for the construction of immunotoxins that would escape preexisting antibodies from vaccination. DT led to FDA approved immunotoxins but anti-drug antibodies (ADA) remain a challenge.

The authors search data banks for DT analogues and showed that toxins (chelonatoxins) from a reptile pathogen could be alternatives to DT. This was demonstrated by retargeting these toxins to alternate receptors expressed by AML or tumor cells.

In addition, the authors show that the T domain from chelonatoxin can translocate

catalytic domain from other toxins more efficiently than DT. Although interesting, the importance of this part seems relative: why change a catalytic toxic domain from the native toxin to that of another toxin affecting small G proteins ? This part should be better explained in the discussion in terms of applicability in biomedicine. Previous experiences showed that most proteins cannot accept the translocation process of DT, limiting applicability of this principle.

Because intracellular delivery of protein-based cargo remains one of the greatest unmet challenges in biotechnology, new targeted platforms that can deliver diverse macromolecules are urgently needed. In our previous publications and patents, we demonstrated that the DT translocase was able to deliver proteins and nucleic acids of a range of sizes and structures efficiently into cells (Auger et al, 2015; Park et al, 2018; Orrell et al. 2020; Arnold et al. 2020, Vidmar et al 2020; 2023; Sugiman-Marangos; 2022). In one case, we showed that delivery of an enzyme, RRSP, that cleaves oncogenic RAS inhibits tumor growth in vivo, demonstrating the great potential of this concept. Given these findings, we were excited to learn that CTT was an even better intracellular delivery platform in some cases and attempted to capture this here to highlight its potential beyond its use as an immunotoxin scaffold.

The authors should also better discuss the comparative potency of the new immunotoxins with respect to the classical DT, which is very powerful in terms of cell toxicity.

In the absence of human serum, DT is as good, or sometimes better (depending on the particular receptor targeting moiety used), but importantly, in all cases, the CT based immunotoxins are much more potent than DT in the presence of human sera. This, is exemplified in Figure 3, and really demonstrates what the potency of each of the tested molecules would be in humans.

Minor: please correct typo line 184

Thanks – done!

5th Jul 2024

Dear Roman,

Thank you for submitting your revised study. We have now received the feedback from referee #2 who evaluated your revised manuscript. As you will see below, this referee is satisfied with the revisions, and I will therefore be able to accept your manuscript once the following points will be addressed:

1/ Manuscript text:

- Please accept previous changes, and only keep in track changes mode any new modification.
- Please provide up to 5 keywords.
- Methods:
 - o Please replace the heading by "Structured Methods".
 - o Please provide a reagents and tools table (listing key reagents, experimental models, software and relevant equipment and including their sources and relevant identifiers) followed by a Methods and Protocols section describing the methods using a step-by-step protocol format. The aim is to facilitate adoption of the methodologies across labs. More information on how to adhere to this format as well as a downloadable template (.docx) for the Reagents and Tools Table can be found in our author guidelines:
<https://www.embopress.org/page/journal/17574684/authorguide#structuredmethods>
 - o Cells: please provide origin, reference, culture conditions, and indicate whether the cells were authenticated and tested for mycoplasma contamination.
 - o Please include a statistic section.
- Data availability: Please remove "All other data generated or analyzed during this study are available in the Appendix"
- Acknowledgements: the information provided in the submission system and in the manuscript should match; please note that Sickkids Research Institute (RI), indicated as a funder in the submission system, is not listed in the acknowledgements. Please also add the projects numbers both in the manuscript and submission system.
- References: they should be placed before the figure legends, and listed in alphabetical order with 10 authors before et al. The DOIs should be removed for published articles.

2/ Figures and Appendix:

- Table EV1: rename inside the file/legend where it is listed as Dataset EV1 - it should be Table EV1
- Appendix: please add page numbers to the table of content. The nomenclature should be corrected to "Appendix Table S1" and "Appendix Figure S1", etc.
- Please make sure that all figures are referenced in the manuscript text. Currently callouts are missing for Fig. 3 and Appendix Fig. S6. There are callouts for Suppl. Table 1: is this Appendix Table S1?
- Please address the queries from our data editors in the figure legends:
 1. Please note that information related to n is missing in the legends of figures 1c-d; 3b; 5d-e.
 2. Although 'n' is provided, please describe the nature of entity for 'n' in the legends of figures 2b-c, e, h; 3d, f; 4c.
 3. Please note that the error bars are not defined in the legends of figures 1d; 5d-e.

3/ Thank you for providing Source Data. Please upload as one file per figure.

4/ Author Checklist:

- Please fill in the section "Cell materials" related to mycoplasma contamination
- Please fill in the entire section "Experimental study design and statistics"
- Please check whether you need to fill the section "Dual Use Research of Concern"

5/ Please include the Paper Explained in the main manuscript text.

6/ Synopsis:

- Thank you for providing a nice synopsis image, please resize it as a jpeg/tiff/png file 550 px wide x 300-600 px high and make sure that the text remains legible.
- I slightly edited your synopsis text to match our style and format, please let me know if you agree with the following or amend as you see fit:

"Pre-existing antibodies against diphtheria toxin (DT) from childhood vaccinations limit the efficacy and widespread use of DT-based immunotoxins. A toxin platform retaining the structural and functional features of DT but not recognized by pre-existing antibodies in human sera was developed.

- Distant DT homologs (ACT1 and ACT2) derived from the reptile pathogen *Austwickia chelonae* were found to have functional translocases and catalytic domains.

- ACT toxins evade recognition by pre-existing antibodies to DT in human sera, showing antibody titers below the limit of quantification.
- Engineered ACT-based immunotoxins retargeted to cancer-associated receptors killed receptor positive cancer cells.
- ACT could represent promising alternative immunotoxin platform for cancer therapy with improved pharmacokinetic and pharmacodynamic properties."

7/ As part of the EMBO Publications transparent editorial process initiative (see our Editorial at <http://embomolmed.embopress.org/content/2/9/329>), EMBO Molecular Medicine will publish online a Review Process File (RPF) to accompany accepted manuscripts.

This file will be published in conjunction with your paper and will include the anonymous referee reports, your point-by-point response and all pertinent correspondence relating to the manuscript. Let us know whether you agree with the publication of the RPF.

I look forward to receiving your revised manuscript.

With kind regards,

Lise

To submit your manuscript, please follow this link:
<https://embomolmed.msubmit.net/cgi-bin/main.plex>

***** Reviewer's comments *****

Referee #2 (Remarks for Author):

The authors addressed most of the comments of this reviewer; the revised manuscript is suitable for publication.

The authors addressed the minor editorial issues.

18th Jul 2024

Dear Roman,

Thank you for submitting your revised files. I am pleased to inform you that your manuscript is accepted for publication and is now being sent to our publisher to be included in the next available issue of EMBO Molecular Medicine!

With kind regards,

Lise
